# Reuse of Water in Laundry Applications with Micro- and Ultrafiltration Ceramic Membrane

**DOI:** 10.3390/membranes12020223

**Published:** 2022-02-15

**Authors:** Beatrice Dal Pio Luogo, Toufic Salim, Wenjing Zhang, Nanna B. Hartmann, Francesca Malpei, Victor M. Candelario

**Affiliations:** 1Politecnico di Milano, Department of Civil and Environmental Engineering (DICA)—Environmental Section, Piazza Leonardo da Vinci 32, 20133 Milano, Italy; bdalpioluogo@gmail.com (B.D.P.L.); francesca.malpei@polimi.it (F.M.); 2Department of Research and Development, LiqTech Ceramics A/S, Industriparken 22C, 2750 Ballerup, Denmark; tsa@liqtech.com; 3Department of Environmental Engineering, Technical University of Denmark, Miljøvej 113, 2800 Kongens Lyngby, Denmark; wenz@env.dtu.dk (W.Z.); nibh@env.dtu.dk (N.B.H.)

**Keywords:** water reuse, industrial laundry, ceramic membrane

## Abstract

This study compares the performance of a microfiltration membrane, made by silicon carbide (SiC) and an ultrafiltration membrane, made by zirconia (ZrO_2_), in the treatment of wastewater from a washing machine designed to clean industrial tents. The filtration of deionized water, containing model microplastics (i.e., nylon fiber), was performed. This was followed by the filtration of real wastewater from a single washing cycle of industrial tents, made from polyvinyl chloride (PVC) textile. The filtration parameters of the membranes and physical-chemical parameters of the wastewater, including the concentration of microplastics in the shape of tent fibers (PVC), were calculated before and after filtration. The microfiltration membrane manifested a greater decrease in permeability (95%) compared to the ultrafiltration membrane (37%). The resulting water quality in terms of Total Solids, turbidity, and microplastics concentration was better for the ultrafiltration. This is evident from 99.2% versus 98.55% removal efficiency of microplastics from the laundering wastewater, respectively.

## 1. Introduction

Water scarcity and water contamination are among the most challenging global issues today. Water reuse plays an important role in solving this challenge. Adding to the conventional chemical pollutants, micro- and nano-plastic (MNPs) are prevalent in the environment, including in aquatic compartments. These contaminants pose an environmental, political, and social concern worldwide because they are ubiquitous and may result in chronic exposure due to their persistency in nature. MNPs can potentially work their way up through food chains in different ecosystems, as a consequence of both aquatic and terrestrial pollution. This, in turn, is partially due to the discharge of inadequately treated wastewater and the application of wastewater sludge on agricultural soils. Microplastics (MPs) can enter the environment readily in the form of MPs, or they can generate in the environment through the fragmentation of larger plastic items. It is estimated that 35% of the total amount of directly released MPs comes from the washing of synthetic textiles [1], making the discharge of MNPs from industrial laundry wastewater one of the biggest sources of MNPs to sewage and surface waters [2,3]. As society moves toward a circular economy by increasing wastewater reuse, there has become an urgent need to remove the MNPs from industrial laundry wastewater to allow for its reuse and to reduce the MNPs load on the receiving wastewater treatment plant (WWTP). However, the current industrial laundries are not capable of removing MNPs on site. An estimated 90% of industrial laundry wastewater discharges to sewage without adequate treatment, contributing to MNPs load in WWTP inlet water [4].

A study conducted by Magni et al. (2018) highlighted that a WWTP in Northern Italy releases 160,000,000 MPs/day to freshwater via treated effluents and 3,400,000,000 MPs/day to sewage sludge that is often reused as fertilizer in agriculture [5]. These numbers pose new issues for the regulation of biosolid disposal in the environment. The washing processes of synthetic textiles have been assessed as the main source of MPs [6,7,8], that enter the oceans [1,3,4,5]. Even if up to 99% of MPs are removed by wastewater treatment plants [9], some, especially the fiber-shaped ones, are released with the effluent [2,10,11] and can pose a threat to ecosystems and human health [3]. For this reason, it is critical to increase the effectiveness of MPs removal at the washing machine discharge before they reach the WWTPs [5,12]. MPs can be separated from laundry wastewater by applying microfiltration membrane technologies. However, they are not capable of removing submicron-sized MNPs, so ultrafiltration could be the solution to remove these small contaminants. 

Membrane-based technologies have been considered as an effective replacement for conventional water and wastewater treatment technologies, such as coagulation, flocculation, advanced oxidation processes, ion exchange, activated carbon adsorption, with the advantage of lower energy consumption, a smaller environmental impact, and efficient separation capability [13,14,15]. To remove MNPs, a microfiltration (MF)/ultrafiltration (UF) membrane-based filtration system can be proposed for laundry wastewater treatment. At present, the UF market is dominated by polymer membranes because of the low production cost. However, these polymer membranes typically suffer from swelling, biofouling, scaling and poor thermal and chemical resistance, which leads to a short lifespan and demands regular cleaning procedures. Besides, they may cause the release of constituent polymers, thus adding MNPs to permeate. On the contrary, ceramic membranes (e.g., SiC, ZrO_2_, and Al_2_O_3_) benefit from higher chemical and physical stability. In particular, SiC is preferred over Al_2_O_3_ due to higher hydrophilicity resulting in high fluxes at low water pressures. This allows the SiC-based filtration process to have a higher capacity and lower fouling potential than other materials [16,17,18,19]. In general, a long lifespan and lower maintenance of ceramic membranes lead to a significant reduction in OPEX and environmental impacts. Hence, ceramic membranes are far superior to polymer membranes for effluents of high temperature, harsh chemical environment, and containing abrasive particles, as seen in industrial laundry wastewater treatment. 

The type of washing process investigated in this study is industrial laundering of tents made of PVC fabric. This material is chosen for being a water-resistant and durable material, making it suitable to cover outdoor structures. The specification of the industrial tent washing machine used in this study, are: 445 kg of tents washed/loaded (on average).Each machine runs on average 3.58 loads/day.

This implied that 1595 kg of tents can be washed per day. If a washing machine runs an average of 4.5 days/week, 7178 kg of tents is washed every week. Assuming that the fiber release behavior of PVC is similar to that of polyester (i.e., assuming a release of 150 mg microplastics/kg textile washed during the first washing cycle), resulting in an average of 1.08 kg/week of PVC microfibers would be released from one single industrial washing machine. If the wastewater does not undergo a proper treatment, addressing MP removal, these could potentially be released into the environment [6]. 

To remove solids from the laundering water, and to permitthe reuse of wastewater in the same washing process, a solution of membrane technologies can be used. 

The present paper focuses on the comparison of two ceramic membranes made by SiC and by ZrO_2_. The membranes are in the micro- and ultra-filtration range, respectively, and are evaluated in terms of MNPs removal and membrane performance. Membranes were tested on a semi/pilot scale under real conditions. Structural characterization of the starting membrane was performed, and the effectiveness of the two membranes was measured by calculating different filtration parameters as critical flux, backflush period, and long-term filtration to evaluate the fouling effect of the membranes. A complete analysis of the feed and permeate water was performed, measuring turbidity, total suspended solids, volatile suspended solids, chemical oxygen demand, conductivity, pH, total solids, total alkalinity, and microplastics.

## 2. Materials and Methods

Two different highly porous multichannel commercial ceramic membranes were used from LiqTech Ceramics A/S, Ballerup, Denmark, which consisted of 30 cylindrical channels with 3 mm diameter for each channel and 305 mm length, with an effective membrane area of 0.09 m2, a microfiltration membrane made by SiC support and SiC membrane (MF) and an ultrafiltration membrane made by SiC support and ZrO2 monoclinic membrane (UF). Membrane characterization was performed with Scanning Electron Microscopy (SEM), FlexSEM 1000 (Hitachi GmbH, Solna, Sweden) and Capillarity Flow Porosimetry (CFP), (3P Instruments, Odelzhausen, Germany) was carried out to measure the mean pore size and pore size distribution of the membranes of three samples from three different locations of the membrane. Before measurement, the pores of the membrane were filled with PorofilTM (fluorinated hydrocarbon) wetting liquid having a surface tension of 16 dyn/cm. The wet curve was obtained by measuring the airflow rate through a sample with increasing pressure. Then, the dry curve was measured by increasing the air pressure using a dry membrane sample. A wet and dry curve was obtained, measuring the air flow rate using the sample with increasing pressure. 

The filtration tests were all conducted on the commercial pilot-scale filtration set-up LiqTech LabBrain (LiqTech Ceramics A/S, Ballerup, Denmark) [17,18]. The unit was run with 1527 L/h of crossflow and with 90% recovery. The retentate and the permeate were connected to the feed tank to perform recirculation. 

Two different sets of experiments were conducted:Filtration of deionized water mixed with nylon microplastics obtained through cryogenic grinding (or cryogrinding) of red nylon fibers with 500 µm length. This technique consists of freezing materials by pouring over liquid nitrogen (−196 °C) and then reducing it into a small particles size through milling (IKA A11 Basic Analytical mill). The length of MPs particles obtained was 80 µm, standard deviation: ±39 µm.Filtration of wastewater from a single washing cycle of a PVC tent in an industrial washing machine.

Deionized water, deionized water plus 0.18 g/L of Nylon fibers of 80 μm and discharge from the first cycle of a washing machine of industrial tent laundering made of poly-vinyl- chloride (PVC) fabric (Hvalsø Teltudlejning ApS, Hvalsø, Denmark) were tested using the LabBrain equipment following the parameters mentioned before. The filtration steps are illustrated in Figure 1. The critical flux measurement was performed by testing with a gradual increase in the permeate flux and the transmembrane pressure was measured (TMP), with a holding time of 15 min in each step to achieve stabilization of the permeate flux and TMP. To control the deposition of solids on the membrane surface, back-flushing (BF) of the membrane was performed, with the permeate feed at 3 bar pressure for 3 s duration. When the back-flushing permeate was pressed back through the membrane by a pump, the maximum interval that should be kept between each back-flush was investigated to achieve an optimal operation, with a stable flux and where only a small permeate consumption is required. To restore the original flux of the membrane, cleaning in place (CIP) of the system was performed with a combination of alkaline, Ultrasil 115 (Ecolab, København, Denmark) that consists of potassium hydroxide (10–20%), sodium hydroxide (10–20%), and ethylenediaminetetraacetate (5–10%) and acids, Ultrasil 75 (Ecolab, København, Denmark), which is a mixture of nitric acid (10–30%) and phosphoric acid (10–30%), following the procedure mentioned by E. Eray et al. [17]. The benefit to industries that use CIP includes faster, less labor-intensive, and more repeatable cleaning, which poses a lower chemical exposure risk. With the right combination of chemicals, it is possible to dissolve as much of the foulant which will then leave the system easily. To confirm the recovery of the membrane, a deionized water permeability was measured after each CIP to be compared with the original one. Once the optimal parameters were obtained, the long-term filtration was performed by setting up the parameters for 96 h.

The feed water and the permeate after the filtration experiments were characterized concerning the main water quality physical-chemical parameters such as pH, conductivity, turbidity, total alkalinity (TAL), total solids (TS), total suspended solids (TSS), volatile suspended solids (VSS), chemical oxygen demand (COD) were calculated with the method illustrated in Table 1. Then, the water was characterized in terms of the presence of microplastics. 

To determine the concentration of MPs, a sample of 20 mL was filtered through a 5 µm pore size filter, and light microscopy was applied to detect microplastics in the sample. Observing the filter at the microscope, the prevalence of particles other than MPs was immediately evident. Hence, a pretreatment with hydrogen peroxide to remove organic matter and improve MP detection. For this purpose, 1 mL H_2_O_2_ was added to a 20 mL sample and incubated at 20 °C. The fibers were imaged and measured using QCapture Pro software; The test was repeated three times, and the fibers were detected through visual counting. The filter was divided into a grid, as shown in Figure 2, which made it easier to count square by square. To speed up the counting, only 12 squares were chosen in a well-distributed manner, each having an area of 0.25 cm^2^. The average number of microplastics counted per square was multiplied by the total number of squares on the filter. The total area of the filter was 23.74 cm^2^, corresponding to 94.94 squares. 

## 3. Results

### 3.1. Membrane Characterization

Figure 3A,C shows the SEM images of the selective layer and Figure 3B,D of the membrane cross-section. As it is possible to see by the results of the two membranes, the UF had a thicker layer and the smaller pore size compared to MF. The layer thickness of MF and UF were ~68 µm and ~87 µm, respectively. Both samples had a homogeneous surface without defects and good adhesion of the grains.

This difference in pore-size range and distribution was also confirmed quantitatively from the results of Capillary Flow Porosimetry, visible in Figure 4 and Table 2. The average values of the maximum d_90_ and d_50_ pore size from the three samples (middle, left, and right part of a tubular membrane) was calculated for both membranes. As it is possible to see from Table 2, 90% of the pores in MF were below 302 nm, while for UF they were below 63 nm, resulting in one order of magnitude of difference between the size of the pores in the two membranes. These tests confirmed that MF is in the microfiltration range and the UF is in the ultrafiltration range, as indicated by the supplier. 

### 3.2. Synthetic Feed

The performance of the two different membranes was studied first in controlled conditions, with a synthetic feed prepared by mixing 0.18 g/L of nylon fibers (80 μm) in deionized water to facilitate the analysis of the microplastic removal studying the behavior of the flux vs. the TMP and analyzing the permeate in both cases.

Prior to the start of the test with the synthetic water, the permeability of the membrane in deionized water was determined. Figure 5 shows the variation of the permeate flux vs. the transmembrane pressure of the deionized water. From the slope of the linear fit of the experimental data, the deionized water permeability can be calculated as ~3400 L/(m^2^·h·bar) for the MF and ~327 L/(m^2^·h·bar) for the UF. The intercept with the y-axis is different from zero in both graphs, and this may be because of the osmotic pressure, of which an initial pressure is necessary to overcome, as well as pressure originated from the pressure drop within the membrane pore [20]. The permeability is also in the range of the micro and ultrafiltration membrane, respectively, for both membranes.

Figure 6 shows permeate flux vs. the transmembrane pressure of the synthetic feed. A loss of flux occurs in the case of MF but not with the UF. In fact, by plotting TMP vs. the flux for the MF and by plotting the tendency line (in orange), which is obtained by interpolating the lower points and represents the behavior that the membrane would have in the case of clean water, it is possible to see that the empirical data, with an increase in the flux, reach a high TMP rapidly and no longer overlaps with the tendency line. This means that the MF undergoes a loss of flux. Hence, it is possible to calculate the critical flux for the MF filtrating small microplastics, which is ~200 L/(m^2^·h). In both cases, the number of microplastics in the permeate was evaluated with a 100% removal.

Even though both membranes presented a full rejection of the microplastics, the presence of the synthetic fibers in the synthetic feed affected the permeability of the microfiltration membranes, leading to the establishment of the critical flux value, differently from the UF membranes. Comparing the size of membrane pores and microplastics particles, it can be assumed that pore-blocking occurred in the MF membrane, possibly generating irreversible long-term fouling. 

### 3.3. Wastewater from Washing Machine of Industrial Tent Laundering

#### 3.3.1. Filtration Experiments

Critical flux is the first parameter calculated to evaluate the performance of the micro- and ultra-filtration membranes with the real feed.

Figure 7 shows the experimental values and the linear fit of the first point describing the behavior of the membrane in case of no loss of flux (as with deionized water flux), and the conjunction of these two lines is the critical flux, which is defined as the threshold flux below which irreversible fouling does not occur, and above which fouling becomes noticeable. In both cases, the tendency line (in red) diverges from the experimental line (blue). 

The critical flux corresponds to the point of separation between the red line (tendency line) and the blue line (empirical data). For MF, the critical flux that results is 100 L/(m^2^·h). As is possible to see in Figure 7A, in the case of UF, the critical flux is 55 L/(m^2^·h), also indicated in Figure 7B.

Once the critical flux is calculated, the backflush interval can be determined. Hence, with a constant value of flux, chosen below the critical flux, the backflush interval was obtained. For the microfiltration membrane, a constant flux of 90 L/(m^2^·h) was set, and for the ultrafiltration membrane, 50 L/(m^2^·h) was set.

For the MF, the initial TMP of 0.19 bar was recovered with a back-flush (BF) time set after 30 min. A second BF was set after 60 min, resulting in a higher TMP compared with the initial one. During the subsequent run of 60 min, the TMP increased and was not recoverable by back-flush. As is visible in Figure 8A, after a BF interval of 1 h, the TMP became too high, and the BF could not revert it to its original state. Therefore, BF for the long term must be set to 20 min, a value just below the BF interval where the TMP can be recovered, to ensure that in long term filtration the permeability with BF without the use of CIP can be recovered. For the UF, after 30 min, the BF could completely recover the initial TMP, which was 0.37 bar, after 90 min of running at a constant flow without BF the TMP was slightly higher, while after a BF interval of 3 h, the TMP became too high, and the BF could not be restored to the original one. Hence, the BF time must be set to 1 h for long-term filtration. 

Because of the larger pores in the microfiltration membrane, the backflush period for MF should be lower than for UF, to avoid irreversible fouling. After determining the critical flux and the back-flush time, the unit was set to perform the long term filtration at optimal conditions. 

The long-term filtration consisted of recirculating the wastewater through the membrane for 4 days, and the permeability, recorded every 30 min, is shown in Figure 9. The settings were based on the previous experiments, with the following parameters: (i) for MF a constant permeate flux of 90 L/(m^2^·h) and backflush every 20 min, (ii) for the UF a constant permeate flux of 50 L/(m^2^·h) and backflush every 60 min. Despite both membranes filtrating at their optimal conditions, the MF underwent a great decrease in permeability of ~95%, while for the UF the permeability loss was less accentuated, at ~37%, for the same period of filtration.

The use of the UF will provide better performance during filtration and less BF during the time, with less permeability loss and avoiding the clogging of the membrane during a longer period. The MF will need to undergo CIP more often than the UF, and the operational cost will be higher in the case of microfiltration membranes.

#### 3.3.2. Physical-Chemical Analysis of Water after Filtration

Table 3 reports the removal efficiencies of the most significant parameters involved in the definition of water quality. The first aspect that is noticeable both qualitatively and quantitatively is the transparency of the permeate, whereby the wastewater feed was very black and dusty, while both permeate streams are clear when observed by the naked eye. This visual improvement is also confirmed by the measurement of turbidity, which changed from 206 NTU of the feed to 10.3 NTU in the MF permeate (and 1.01 NTU in the UF permeate. It can be noticed that the pH did not change to a significant degree after filtration. In the feed, it was 6.8, and for MF permeate, it remained around neutrality (7.06), while for the UF permeate, it became slightly alkaline (8.36), as can be easily noticed by observing the Total Alkalinity of the feed (5.77 meq/L) and of the UF permeate (8.53 meq/L), which is higher, while TAL of the MF permeate was 6.81 meq/L, this change can be due to the cleaning process. Furthermore, the UF membrane could absorb some alkalinity from the CIP and increase the pH slightly. Conductivity remains more or less the same before and after filtration, around 800 μS/cm. This could be predictable since the membrane is not capable of retaining salts. What presents a significant variation, as expected, is the solid matter contained in the wastewater, especially in terms of TSS and VSS. Total suspended solids were 221.67 mg/L in the feed, while after filtration with MF they become 52.5 mg/L (76.3% efficiency of removal), and with UF they become 9 mg/L (95.9% removal). This difference in the removal efficiency is explained by the physical separation mechanism performed by the membranes. In the case of UF, it has smaller pores on its selective layer, at 50% below 42.6 nm, so it is capable of removing more solids than the MF, which has larger pores (at 50% below 247.2 nm). Focusing on the Total dissolved solids, in the wastewater, a negligible reduction was observed for MF filtration (feed: 737 mg/L, permeate: 731 mg/L), and a 20% reduction in concentration was observed for UF filtration. The reduction of TDS can be explained by the higher capability of the UF membrane to retain organic and inorganic substances in colloidal forms. Further treatment, such as Reverse Osmosis (RO) or Forward Osmosis (FO), could be required to reduce TDS. Regarding the organic matter contained in the water, Volatile Suspended Solids changed from 121.67 mg/L in the feed to 25 mg/L in the MF permeate (the removal occurred with 79% efficiency) while they are negligible in the UF permeate. The Chemical Oxygen Demand in the wastewater feed was 424 mg/L, while in the MF permeate, it was 84.3 mg/L, and in the UF permeate, it was 68.7 mg/L. In general, the UF membrane achieved a better removal of Turbidity, TSS, VSS, and COD, because of its small pores in its selective layer. 

The characterization of the water quality then focused on the microplastics, by calculating the average number of microplastics found in 20 mL of wastewater feed by visual counting through light microscopy. The number of microplastics found can be approximated to around 900 microplastics for 20 mL, thus corresponding to 45,000 microplastics/L. Supposing the average volume of 1 single microplastic to be 353 × 10^−9^ cm^3^ (based on an average length of 0.5 mm, a radius of 0.015 mm, and a cylindric geometry), and assuming a density of PVC in the range 1.40–1.45 g/cm^3^, the weight of a single microplastic can be approximated to 503 × 10^−9^ g. Therefore, the concentration in the mass of MPs in the wastewater feed would correspond to 22.6 mg/L. This is a value comparable to 20.5 mg/L of fibers, released after the 1st washing cycle according to a previous study [1]. After filtration, the number of microplastics found in the permeate from filtration with the MF membrane was 1250 microplastics/L, with a concentration of 0.3 mg/L, meaning that the microfiltration membrane was able to remove 98.5% of microplastic particles from the laundering wastewater. A similar calculation was performed for the microplastics found in the permeate from the filtration with the UF membrane, resulting in 450 microplastics/L, with a concentration of 0.135 mg/L. This value is very low and means that the percentage of removal of microplastics for this membrane is around 99.2%. In both cases, almost all microplastics have been removed, with higher efficiency of the UF membrane, probably because of the smaller pore size.

## 4. Conclusions

In the present work, a comparison of two ceramic membranes (MF and UF) was performed, with the aim to (1) evaluate their capability to retain MPs and (2) to evaluate the effect of the presence of MPS on membrane fouling. Tests were conducted with a synthetic feed (deionized water + MPs) and with real wastewater from the 1st washing cycle of an industrial laundry application. The key findings are summarized as follows:From the membrane characterization, it was found that both membranes had a defect-free and homogeneous surface. Furthermore, the membrane made by SiC was in the microfiltration range, with d_90_ of ~302 nm, whereas the membrane made using ZrO_2_ was in the ultrafiltration range, with a d_90_ of ~52 nm.The filtration of the synthetic feed with nylon fibers of 80 µm showed a critical flux value, in the case of MF, of 200 L/(m^2^·h). This demonstrates an effect of MPs in terms, most probably, of pore blocking. With the capabilities of the unit, it was not possible to obtain a critical flux for the UF, because no reduction of flux was observed along with the increasing TMP cycles. There is a clear indication that the fouling occurs earlier in MF compared to UF. In both cases, a 100% rate of removal of the fibers was achieved.With the filtration of the real wastewater from the tent laundry outlet, the critical flux value and backflush period for the MF was 90 L/(m^2^·h) with a 20 min period and 50 L/(m^2^·h) and 60 min period for the UF. After 4 days of constant filtration, there was a considerable decrease in the permeability of MF (~95%), while much smaller in the case of UF (~37%). Therefore, the better performance of UF in real applications can be established, with a lower necessity of CIP and longer operation periods.Based on the water analysis of the feed and permeate during the long-term filtration of both membranes, we can conclude that the UF results in better water quality in the permeate compared to the MF, with almost a 100% rate of removal in all studied parameters. The results obtained in this study pave the way for future wastewater treatment systems for industrial laundries, where the UF ceramic membranes can be used as the polishing step to remove MPs before being reused or discharged into the municipal wastewater stream.

## Figures and Tables

**Figure 1 membranes-12-00223-f001:**
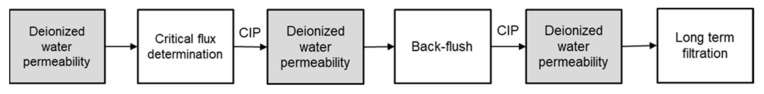
Filtration test plan.

**Figure 2 membranes-12-00223-f002:**
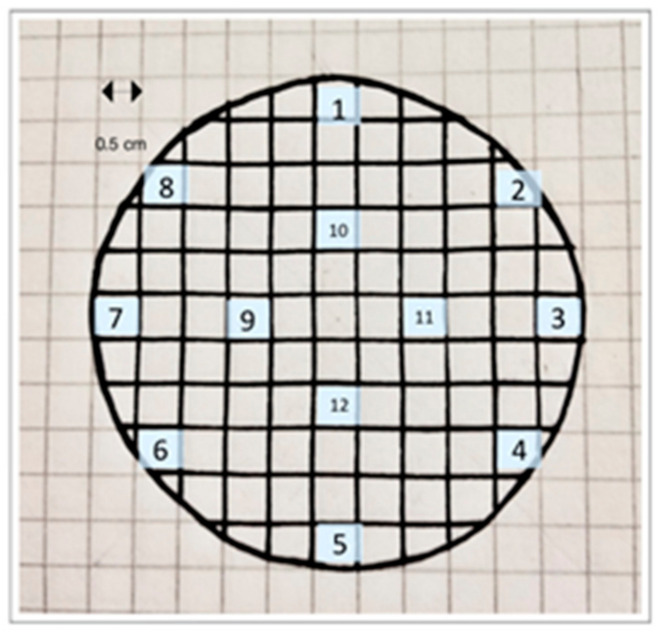
Scheme used for the visual counting of microplastics.

**Figure 3 membranes-12-00223-f003:**
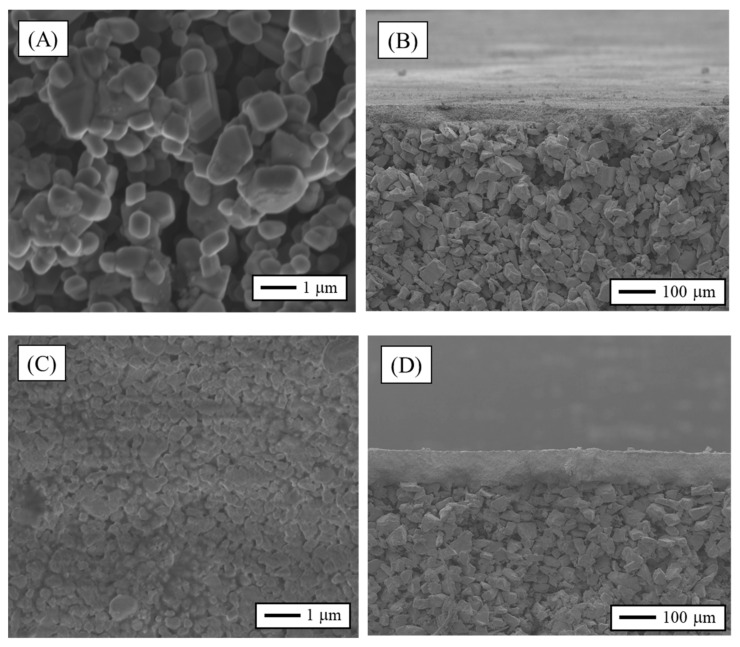
Results of SEM, selective layer (**A**) MF (**C**) UF, cross-section (**B**) MF (**D**) UF.

**Figure 4 membranes-12-00223-f004:**
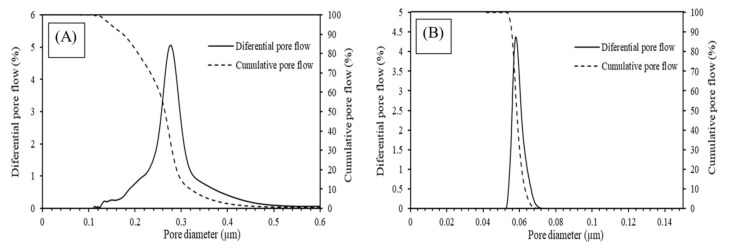
Results of the pore-size distribution of (**A**) MF (**B**) UF.

**Figure 5 membranes-12-00223-f005:**
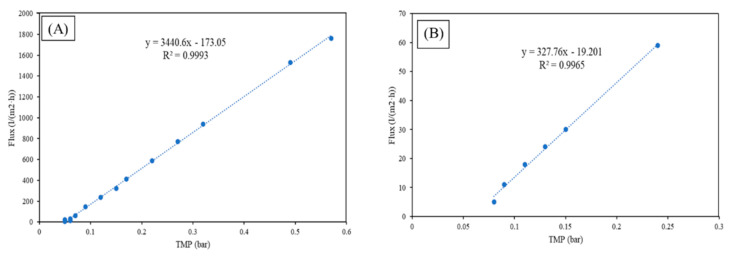
Permeate flux vs. transmembrane pressure of the deionized water for (**A**) MF and (**B**) UF.

**Figure 6 membranes-12-00223-f006:**
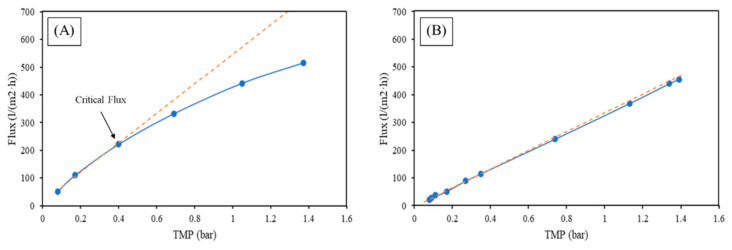
Filtration test with 0.18 g/L small microplastics (**A**) MF (**B**) UF. Where the orange line is theoretical for deionized water.

**Figure 7 membranes-12-00223-f007:**
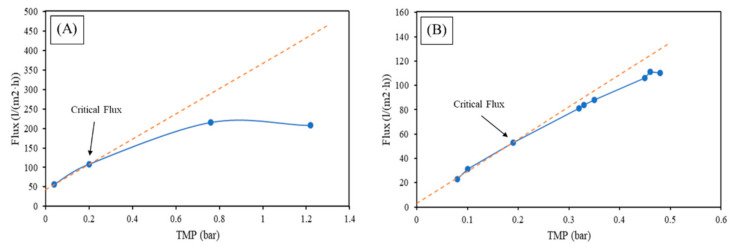
Critical flux determination of feed from the first cycle of a washing machine of industrial tent laundering (**A**) MF (**B**) UF. Where the orange line is theoretical for deionized water.

**Figure 8 membranes-12-00223-f008:**
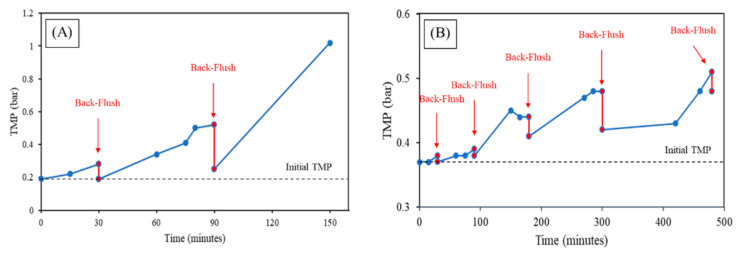
Back-flush time determination (**A**) MF and (**B**) UF.

**Figure 9 membranes-12-00223-f009:**
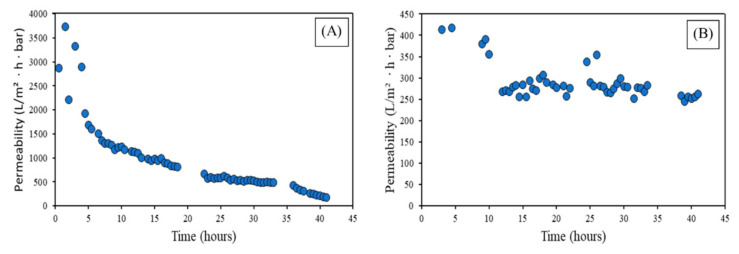
Long-term experiment, permeability vs. time (**A**) MF (**B**) UF.

**Table 1 membranes-12-00223-t001:** Water characterization parameters and instruments used.

Parameter	Unit	Instrument/Method
pH	-	pH probe HQ40D (Hach, Loveland, CO, USA)
Turbidity	NTU	Turbidimeter TN-100 (Thermo Scientific Eu, Bufalo, NY, USA)
Conductivity	uS/cm	Conductivity meter EC400 model ExStik
TDS	mg/L	Weighting and drying filter at 105 °C
TSS	mg/L	Weighting and drying water sample at 105 °C
VSS	mg/L	Weighting and drying water sample at 500 °C
TAL	mg/L	Titration with sulfuric acid
COD	mg/L	Cuvette test for COD, 15–150 mg/L O_2_

**Table 2 membranes-12-00223-t002:** Results of Capillarity Flow Porosimetry.

Membrane	Maximum Pore Size (nm)	d90 (nm)	d50 (nm)
MF	604	302	247
UF	74	63	58

**Table 3 membranes-12-00223-t003:** The removal efficiency of the main wastewater parameters.

Removal Efficiency (%)		
Parameter	MF	UF
Turbidity	95	99.5
TSS	76.3	95.9
VSS	79	100
COD	80	83.8
microplastics	98.5	99.2

## Data Availability

Not applicable.

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
