# Peer review of "Reuse of Water in Laundry Applications with Micro- and Ultrafiltration Ceramic Membrane"

_membranes, 2022, doi:10.3390/membranes12020223_

Round 1
Reviewer 1 Report
The reuse of laundry wastewater by microfiltration and nanofiltration was investigated. The presence of radical species has been evaluated. In addition, the filtration of artificial water containing microplastics was evaluated. Globally, the experimental procedures and results are properly presented but several discussion statements are rather weak. Some issues have to be addressed:
- Line 45. The abbreviation MPs must be defined here instead of in line 48.
- Lines 49-51. References 6, 7 and 8 seems to be missed in the text.
- Some experiments have been performed with artificial water containing microplastics. The main characteristics of this water must be provided.
- Table 1. Analysis of solid fractions has to be reviewed. Is the analytical method for TSS and VSS exactly the same?.
- Figure 5. The intercept on the y-axis is far from 0 in both graphs. An explanation must be provided in the manuscript.
- Line 204. “…plotting TMP vs Flux for the MF…” Where is this plot in the manuscript?.
- Line 275-300. The main quality parameters for feed and permeate stream could be summarized in a Table, together with the removal efficiencies.
- Most of the figures are rather small and it is difficult to read the legends and distinguish the symbols.
- The paper contains some typographical errors. Please read carefully and correct.
Author Response
The authors really appreciate the reviewer’s comments on the manuscript.
All the changes are addressed in the manuscript marked by yellow, the text was modified to find and correct typographical errors, and to be more understanding, and all figures are made bigger.
Please check the attachment.

Reviewer 2 Report
The paper is written like a research report rather than a scientific article, which presents and explains obtained results. This should be corrected and completed.
All notes and comments are marked in the attached manuscript file.

Author Response
The authors really appreciate the reviewer’s comments on the manuscript.
All the changes are addressed in the manuscript marked by yellow, the text was modified to find and correct typographical errors, and to be more understanding.
Please check the attachment.

Reviewer 3 Report
A paper entitled “Reuse of water in laundry applications with micro and ultra-filtration ceramic membrane” conducted the MF and UF treatment using artificial wastewater including nylon fiber as a nano-plastics. The experimental results revealed the critical water flux which start to cause membrane fouling, and subsequent rejection performance using MF and UF membranes. This test will be quite important for removing micro- and nano-plastic for laundry wastewater.
However, in this paper, a deep discussion is crucially lacked and not suitable as an academic paper to be published in Membrane journal (IF = 3). This paper just showed the results and then introduce the data with the reasonable explanation using some reference. Therefore, this paper should be dramatically revised for re-submitting the journal, such as deep theoretical discussion, experimental design and adequate writing design for easy understanding.
To improve the paper quality from academic viewpoint, the following comments should be considered, at least.
(C1) Membrane module
Please add the effective membrane area of module used in this study.
(C2) Nylon microplastic
Please add the SEM image of nylon microplastic.
(C3) “Length and 118 color: 80 um (red)”, and “Nylon fibers of 0.08 μm”
From this information, the nylon fiber has 80 um length and 80 nm diameter? Is this correct?
(C4) Critical flux which start to cause fouling
Critical flux which starts to cause fouling in Figure 6 should also depend on the total filtration volume, how long time did the authors conducted this test? In addition, for comparison, please change the y-axis value of Figures 6 to same.
(C5) Data for figure 6 and figure 7
Experimental procedure for figure 6 and figure 7 seems to be lacked, and therefore, it was difficult to understand these results. Please add more detail in the experimental section.
(C6) Deep discussion is lacked
In this study, there are no deep discussion. Please add the important point of this study such as novelty, new finding and important aspect. This will be quite important to be accepted in Membrane journal which has impact factor about 3.
Author Response
The authors really appreciate the reviewer’s comments on the manuscript.
All the changes are addressed in the manuscript marked by yellow, the text was modified to find and correct typographical errors, and to be more understanding, and all figures are made bigger.
(C2) Nylon microplastic
Please add the SEM image of nylon microplastic.
Response: Due to the size of the microplastic ( 80µm) the authors consider that it is not necessary to add SEM image of the microplastic.
Please check the attachment.

Round 2
Reviewer 1 Report
Most of the recommendations made by the Reviewer have been considered and the paper can be accepted for publication.
Author Response
Thank you for your time.
Reviewer 2 Report
The whole text needs stylistic and linguistic correction.
Lines 271-280 - Figure 9 shows changes in permeability rather than TMP.
Lines 146-148 - The text lacks information supporting the effectiveness of CIP in permeability recovery.
Author Response
Thank you so much for your comments, attached you have the response letter.

Reviewer 3 Report
Thank you very much for your revision. I have an additional comment as follows:
(C1) “The intercept with y-axis is different from zero in both graphs, and this can be because of the osmotic pressure which an initial pressure is necessary overcome [21].”
Because the authors used deionized water, the osmotic pressure of the solution should be zero. Therefore, the above explain will be not correct. I think the reason is due to the entry pressure originated from pressure drop within the membrane pore, so that please consider again.
Author Response
Thank you so much for your review, we added this sentence to the manuscript.